# Uneven Levels of 5S and 45S rDNA Site Number and Loci Variations across Wild *Chrysanthemum* Accessions

**DOI:** 10.3390/genes13050894

**Published:** 2022-05-17

**Authors:** Jun He, Yong Zhao, Shuangshuang Zhang, Yanze He, Jiafu Jiang, Sumei Chen, Weimin Fang, Zhiyong Guan, Yuan Liao, Zhenxing Wang, Fadi Chen, Haibin Wang

**Affiliations:** State Key Laboratory of Crop Genetics and Germplasm Enhancement, Key Laboratory of Landscaping, Ministry of Agriculture and Rural Affairs, Key Laboratory of Biology of Ornamental Plants in East China, National Forestry and Grassland Administration, College of Horticulture, Nanjing Agricultural University, Nanjing 210095, China; 2020204041@stu.njau.edu.cn (J.H.); zy@stu.njau.edu.cn (Y.Z.); 2021104106@stu.edu.cn (S.Z.); 2021104088@stu.njau.edu.cn (Y.H.); jiangjiafu@njau.edu.cn (J.J.); chensm@njau.edu.cn (S.C.); fangwm@njau.edu.cn (W.F.); guanzhy@njau.edu.cn (Z.G.); liaoyuan@njau.edu.cn (Y.L.); wangzx@njau.edu.cn (Z.W.); chenfd@njau.edu.cn (F.C.)

**Keywords:** *Chrysanthemum*, Asteraceae, 5S rDNA, 45S rDNA, oligonucleotide fluorescence in situ hybridization (Oligo-FISH)

## Abstract

Ribosomal DNA (rDNA) is an excellent cytogenetic marker owing to its tandem arrangement and high copy numbers. However, comparative studies have focused more on the number of rDNA site variations within the *Chrysanthemum* genus, and studies on the types of rDNA sites with the same experimental procedures at the species levels are lacking. To further explore the number and types of rDNA site variations, we combined related data to draw ideograms of the rDNA sites of *Chrysanthemum* accessions using oligonucleotide fluorescence in situ hybridization (Oligo-FISH). Latent variations (such as polymorphisms of 45S rDNA sites and co-localized 5S-45S rDNA) also occurred among the investigated accessions. Meanwhile, a significant correlation was observed between the number of 5S rDNA sites and chromosome number. Additionally, the clumped and concentrated geographical distribution of different ploidy *Chrysanthemum* accessions may significantly promote the karyotype evolution. Based on the results above, we identified the formation mechanism of rDNA variations. Furthermore, these findings may provide a reliable method to examine the sites and number of rDNA variations among *Chrysanthemum* and its related accessions and allow researchers to further understand the evolutionary and phylogenetic relationships of the *Chrysanthemum* genus.

## 1. Introduction

Asteraceae (Compositae) is one of the most prominent angiosperm families in herbaceous plants worldwide, containing 1600 genera [1]. Significantly, the Anthemideae tribe belonging to Asteraceae is a large and widespread group of flowering plants mainly distributed in Central Asia, the Mediterranean Basin, and Southern Africa [2]. Especially the cultivated chrysanthemums among the *Chrysanthemum* genus (Asteraceae, Anthemideae tribe), as some of the most widely marketed cut flowers globally, are popular owing to their wide variety of flower variations in size, color, and shape; long vase-life; early production; and photoperiod responses [3]. In addition, other species in the *Chrysanthemum* genus, primarily native to Asia, are essential germplasm resources for cultivated chrysanthemum (*Chrysanthemum morifolium*) and contribute to increasing its aesthetic value [4,5]. Consequently, promoting studies on the origin and molecular cytogenetics of wild *Chrysanthemum* accessions may further aid the understanding of phylogenetic and species relationships [6].

Repetitive DNA occupies a significant fraction of the nuclear genome in higher eukaryotes and contributes significantly to the plant chromosome structure [7,8]. Turnover of this repetitive DNA occurs across the genome with repetitive DNA variations, such as amplification, deletion, transposition, and mutation. This may be associated with population divergence and speciation processes. Ribosomal DNA (rDNA), a type of repetitive DNA, is highly conserved and ubiquitously present in plants under natural selection pressure [9]. The rDNA has two distinct gene classes in higher eukaryotes: the primary 45S rDNA cluster encoding 28S (26S), 5.8S, and 18S rRNA and the minor 5S rDNA cluster encoding 5S rRNA [4,10,11,12,13].

Consequently, 5S rDNA and 45S rDNA have been shown to be good cytogenetic markers by tandemly arranging and presenting in high copy numbers (up to several thousand copies) with different chromosomal distributions [14,15]. In addition, 5S rDNA and 45S rDNA are regarded as vital markers for fluorescence in situ hybridization (FISH) analysis in molecular phylogenetics and evolution. They are widely used in cytogenetic research, not only for essential crops [16,17], but also for horticultural plants, such as strawberry [18], cucumber [19], sweet potato [20], and chrysanthemum [4,11,21]. For example, the rDNA-FISH results for *Taraxacum* (Asteraceae) suggest a highly dynamic evolution of karyotypes among the genera owing to its polyploid apomictic events [22]. In the *Rosa* genus, rDNA probes were used to observe meiotic chromosome behavior and further the retention of substantial ancient rDNA sequences in genomes resonates with drastic allelic heterozygosity encountered in previous studies [23].

Compared with traditional FISH analysis, the novel oligonucleotide FISH (Oligo-FISH) technology is extensively utilized in related cytogenetic research because of its cost-effectiveness [24,25,26]. Moreover, based on high-throughput DNA sequencing data, oligonucleotide probes have been developed and successfully applied to many plants [27]. Therefore, examining the rDNA sites by Oligo-FISH would efficiently contribute to further cytogenetic research of the *Chrysanthemum* genus.

In a previous study, the amplification or deletion of rDNA arrays during the evolutionary process was observed, which represented the inter-individual fluctuations in several plant groups [28,29]. Studies have assessed the extent of rDNA variation at the sequence, site, and copy number levels in some accessions of the *Chrysanthemum* genus [2,3,4,6,21]. In addition, it has been previously reported that 45S rDNA sites may display intra-specific variations in size and number, and most variations are restricted to artificial hybrids, polyploid systems, and crop species [30,31]. These findings further illustrate the possibility for rDNA variations. However, comparative studies have focused more on rDNA site number variations among the *Chrysanthemum* genus [4,12,15,21,32,33] and explored the types of rDNA sites with the same experimental procedures at accession levels. Crucially, the 5S rDNA and pTa71 (45S rDNA) probes in the conventional FISH analysis contain some heterologous fragments, and non-specific sequences may result in background noise and interfere with the FISH results. In a previous study, prelabeled oligomer probes of 5S and 45S rDNA were designed based on the reference genome data of *Chrysanthemum* and its allied genera in the Asteraceae, and more credible FISH results were obtained [5].

In this study, we aimed to further explore the 5S and 45S rDNA sites by conducting the Oligo-FISH analysis among ten wild *Chrysanthemum* plants ranging from 2n = 18 (2x) to 2n = 54 (6x). Consequently, our specific aims were to (i) clarify the number and different types of 5S and 45S rDNA sites, (ii) analyze the number of rDNA sites associated with geographic distribution and ploidy, and (iii) identify the possible underlying mechanisms for the variation in rDNA sites.

## 2. Materials and Methods

### 2.1. Plant Materials

*Chrysanthemum* plants from diploid (2n = 2x = 18) to hexaploid (2n = 6x = 54) lines were maintained at the Chrysanthemum Germplasm Resource Preserving Center, Nanjing Agricultural University, China (32°05′ N, 118°8′ E, 58 m elevation) (Table 1). Ten investigated plant materials in this study were identified using cytogenetics and morphological analysis. Morphological identification of leaves and inflorescences (ray flowers and tubular flowers) was conducted separately during the periods of vegetative growth and blossoming based on the morphological descriptions in Flora of China. All plants were propagated through cuttings, and the medium contained a 2:2:1 (*v*/*v*) mixture of perlite, vermiculite, and leaf mold. Rooted seedlings were grown in a greenhouse at 22 °C during the day and at least 15 °C at night, with a relative humidity range of 70–75% under natural light.

### 2.2. Cytological Preparation and Multiplex Oligo-FISH

The chromosome spread preparations and Oligo-FISH analysis were performed as previously described [34,35,36] with minor modifications in chromosome preparations. The root tips from cutting seedlings measuring about 2 cm were taken, pretreated in 0.2 μmol/L amiprophos-methyl solutions (APM A0185, DuchefaBiochemie, The Netherlands) for three hours, and washed thrice using tap water. Next, the roots were excised, placed in 1.5 mL Eppendorf tubes, and treated with N_2_O at 1.0 MPa for one hour. The root tips were transferred to 90% acetic acid for 10 min on an icebox and then stored in 70% ethanol at −20 °C until their use in chromosome preparations. After cell spreads were dried on slides, then they were subjected to UV-crosslinked treatment (total energy, 120 mJ/cm^2^). Then, at the center of the cell spreads, 10.0 μL hybridization solution per spread containing 1.0 μL 5S rDNA probe (0.55 ng/μL), 1.0 μL 45S rDNA probe (0.55 ng/μL), and 8.0 μL buffer (equal amount of 1 × TE and 2 × SSC) was dropped. After the application of a plastic coverslip, the slide preparation was denatured by being placed on a wet paper towel in an aluminum tray floating in boiling water (100 °C) for 5 min in dark conditions. Finally, the slides were incubated at 55 °C overnight in a humidity chamber containing 2 × SSC soaked paper toweling. The next day, slides were washed in 2 × SSC for 5 min at room temperature. After drying, the slides were mounted with DAPI mounting medium (H-1200, Vector Laboratories, Burlingame, CA, USA) while we waited to make observations.

The oligonucleotide probes of 5S and 45S rDNA were developed according to the method described by Waminal et al. [37] and then synthesized by General Biosystems (Chuzhou, Anhui province, China), and the sequences were as previously described by He et al. [5]. Finally, the chromosomes were visualized with an Olympus BX60 microscope (Olympus, Tokyo, Japan). All synthetic oligonucleotide probes for 5S rDNA and 45S rDNA emerged as distinct fluorescent signals at a 0.55 ng/μL concentration.

### 2.3. Karyotyping Analysis

Images were captured using a SPOT CCD camera (SPOT Cooled Color Digital, Olympus DP72, Tokyo, Japan). Then, multi-color component images were merged using Cellsens Dimension software (version 1.6). For karyotyping, 3–5 cells from each accession were observed, and karyotypes were generally obtained from a single cell. Otherwise, they were sampled from 1 to 4 cells because of overlapping chromosomes.

Furthermore, chromosome measurements based on Oligo-FISH results and idiograms were obtained using the computer software KaryoMeasure [38]. Finally, the idiograms of the investigated materials were integrated using Photoshop (2021) (Adobe, San Jose, CA, USA).

## 3. Results

### 3.1. Chromosome Numbers and Karyotype Features

Ten wild *Chrysanthemum* plants were analyzed in this study. We identified ten wild *Chrysanthemum* plants by conducting a morphological analysis and cytological observations (Figure 1). These results suggest that the chromosome numbers ranged from 2n = 18 (2x) to 2n = 54 (6x), following previous reports. In addition, the karyotypes of *Chrysanthemum* accessions mostly comprised metacentric and submetacentric chromosomes of similar sizes, and the B chromosome rarely emerged in *C. indicum* (NAU079, Taibaishan, Shaanxi, China) (Figure 1h).

### 3.2. Geographic Distribution of Wild Chrysanthemum Genus Plants

The geographic distribution plays a pivotal role in the karyotype evolution of the *Chrysanthemum* genus. We visualized the sources of related wild *Chrysanthemum* plants listed in Table 1 using RStudio (Version 1.2.5033) (Figure 2). 

From the result of the geographic distribution analysis, most *Chrysanthemum* accessions were located in the mid-latitude area of China (Figure 2a). In addition, the diploid and tetraploid *Chrysanthemum* accessions were distributed along the Yangtze River and Yellow River, which may make the dispersion of seeds and pollens by water possible (Figure 2b). Moreover, the detailed topographic amplitude map equally supports the spreading rules of seeds and pollens (Figure 2c). With the spread of seeds and pollens, different ploidy *Chrysanthemum* accessions simultaneously gathered together to form diverse populations, extensively promoting karyotype evolution among the *Chrysanthemum* genus.

### 3.3. Identification of 5S and 45S rDNA Sites in Chrysanthemum Accessions Using Oligo-FISH

The double-color Oligo-FISH localization of the 5S and 45S rDNA loci on mitotic metaphase chromosomes was assessed for the wild *Chrysanthemum* accessions listed in Table 1 (Figure 3). Overall, the numbers and positions of 5S rDNA sites were relatively conserved among all examined *Chrysanthemum* accessions, unlike that of 45S rDNA, which varied considerably in the number and parts of its sites.

For 45S rDNA, the number of sites ranged from 3 (*C. lavandulifolium* var. *aromaticum* (NAU010) and *C. mongolicum* (NAU011)) to 17 in an autotetraploid *C. nankingense* (NAU172) (Figure 3). Additionally, the Oligo-FISH results indicated that 45S rDNA sites were mainly located at chromosomal termini and minor sites were located at the interstitial region of chromosomes. Simultaneously, the signal intensity levels of the *Chrysanthemum* accessions showed substantial differences.

Compared to 45S rDNA, the number of 5S rDNA sites was conserved across all ten *Chrysanthemum* accessions and may be related to its ploidy, except for the diploid plants *C. indicum* (Wuhan, Hubei) (NAU031) (Figure 3a) and *C. mongolicum* (NAU011) (Figure 3d). Compared to hexaploid wild *Chrysanthemum* accessions, the diploid and tetraploid wild *Chrysanthemum* accessions account for a large proportion of the wild *Chrysanthemum* germplasm. Consequently, we conducted the t-test analysis mainly based on the diploid and tetraploid wild *Chrysanthemum* accessions. As shown in Table 1, we further found that a significant correlation existed between the number of 5S rDNA sites and chromosomes (*p* < 0.001), but not between 45S rDNA sites and chromosome numbers (*p* > 0.05) (Figure 4). In addition, we found that the 5S rDNA signals of ten *Chrysanthemum* accessions were mainly located in subterminal chromosome regions (interstitial regions near centromeres). However, 5S rDNA signals emerged at the chromosomal termini and interstitial areas near the terminal in diploid *C. mongolicum* (NAU011) and *C. indicum* (Wuhan, Hubei) (NAU031), respectively. Consequently, we hypothesized potential chromosomal variations in the two accessions.

Significantly, the 5S rDNA and 45S rDNA signals were colocalized on the same chromosome in diploid *C. indicum* (Wuhan, Hubei) (NAU031) (Figure 3a) and tetraploid *C. indicum* (Tianzhushan, Shaanxi) (NAU047) (Figure 3f).

### 3.4. Patterns of rDNA Sites

Consistently, by integrating the previous related data (Table 1), we constructed the standard karyotype and painted the idiograms using the computer software KaryoMeasure, then the rDNA sites were marked (Figure 5). Based on the results shown above, we divided the patterns of 5S rDNA and 45S rDNA sites into three types.

For 5S rDNA sites, there were three basic types: type I (terminal), type II (interstitial on the short arm), and type III (interstitial on the long arm). Statistically, the 5S rDNA sites were more likely to be located at the subterminal chromosome regions (type I and type II) rather than at chromosomal termini (Type III). Likewise, statistical data further demonstrated that 5S rDNA sites were conserved across *Chrysanthemum* accessions (Table 1).

The patterns found in the 45S rDNA sites were divided into three types: type I (terminal on the short arm), type II (terminal on the long arm), and type III (interstitial). They indicated that most 45S rDNA sites were detected on chromosomal termini (type I and type II), which coincided with previously reported results. However, interstitial areas (type III) of 45S rDNA rarely appeared in *C. indicum* (Botanical Garden, Beijing) (NAU077) and *C. nankingense* (Nanjing, Jiangsu) (NAU172). Briefly, all of the above findings proved that unequal crossing-over or ectopic recombination might emerge in 45S rDNA sites, leading to a highly variable number of 45S rDNA sites.

Furthermore, 5S rDNA and 45S rDNA signals were colocalized on the same chromosome of three accessions, *C. indicum* (Shennongjia, Hubei) (NAU030), *C. indicum* (Wuhan, Hubei) (NAU031), and *C. indicum* (Tianzhushan, Shaanxi) (NAU047), which has rarely been reported in previous studies. Hence, we inferred that these three accessions were highly heterogeneous and exhibited potential chromosomal variation among the 18 investigated accessions.

## 4. Discussion

### 4.1. Distribution Patterns of rDNA Sites in Wild Chrysanthemum Genus Plants 

Typically, in the prevailing views, the highly repetitive sequences constitute a significant fraction of the highest plant genomes (from 20% to 90%) and are arranged hundreds or thousands of times over in separate arrays [8,39,40]. However, the specific functions of these repetitive sequences in the genomes are not yet completely understood. With the improvement of high-throughput DNA sequencing techniques, repetitive DNA sequences have been identified using repetitive sequence analysis software. These repetitive sequences seem to be associated with the organization, evolution, and behavior of plant genomes [41]. 

The rDNA has been shown to be an excellent cytogenetic marker in related studies [42]. In the genomes of higher eukaryotes, rDNA comprises 45S and 5S rDNA. The 45S rDNA cluster is located in the nucleolar organizer region and encodes the 28S (26S), 5.8S, and 18S rRNAs [16,43]. Nevertheless, the 5S rDNA cluster embodies a highly conserved coding region of approximately 120 bp, even among unrelated species, and a non-transcribed spacer region that varies between 100 and 900 bp [20]. Based on the sequences of 5S and 45S rDNA, genome restructuring [44], intra-species genetic diversity [31], and re-discovery of the status of certain species [45] can be easily visualized using FISH analysis. In previous studies, traditional plasmid DNA probes of rDNA (such as pTa71, consisting of a 9-kb *EcoRI* fragment of rDNA derived from *Triticum aestivum*) were used to observe the distribution of rDNA sites in the *Chrysanthemum* genus and its allied genera in the Asteraceae [11,32,33]. With the development of DNA synthesis technology, novel Oligo-FISH technology has been widely utilized instead of conventional FISH technology [37]. Consequently, oligo probes of rDNA have been employed in cytogenetic studies of the *Chrysanthemum* genus [2,3,4,6,21]. Nonetheless, the results of the FISH analysis were limited to karyotype and statistical investigations of sites, and the patterns of rDNA sites with the same experimental procedures at accession levels are lacking. Therefore, examining the differences in the number and distribution patterns of rDNA sites can help to further analyze the chromosomal behavior of different species within the genera.

In this study, we integrated the previously reported data listed in Table 1 to paint idiograms of the rDNA sites of *Chrysanthemum* mitotic metaphase chromosomes and divided the patterns of 5S rDNA and 45S rDNA sites into three equal types (Figure 5). For 5S rDNA sites, there are three basic types: type I (terminal), type II (interstitial on the short arm), and type III (interstitial on the long arm). These results further demonstrated that 5S rDNA sites were conserved across *Chrysanthemum* accessions. The patterns of the 45S rDNA sites were divided into three types: type I (terminal on the short arm), type II (terminal on the long arm), and type III (interstitial). Coinciding with the results previously reported, a highly variable number of 45S rDNA sites was observed.

Generally, 5S and 45S rDNA sites tend to be distributed independently on chromosomes owing to physical distance [46]. Although the probability of 5S-45S rDNA colocalization is relatively low, this phenomenon does exist in some genera, such as *Brassica* [47], *Cucumis* [48], and *Artemisia* [49]. As reported for another genus, we also found that the 5S rDNA and 45S rDNA signals were colocalized on the same chromosome of three accessions of the *Chrysanthemum* genus: *C. indicum* (Shennongjia, Hubei, China) (NAU030), *C. indicum* (Wuhan, Hubei, China) (NAU031), and *C. indicum* (Tianzhushan, Shaanxi, China) (NAU047) (Figure 3 and Figure 4). Consequently, the composition of the polyploid *Chrysanthemum* genome is considered very complex now. Although the potential chromosomal variations, such as translocation or inversion, may not involve a loss or addition of chromosome materials in the *Chrysanthemum* genus, these variations frequently become associated with differences, duplications, and unbalanced combinations of genetic units [50]. These results further indicate that potential chromosomal variation might occur not only in polyploid *Chrysanthemum* species and F1 hybrids, but also in diploid species. Finally, these variations might also be facilitated by transposable elements and important ornamental and resistance characters during the evolution of the species in *Chrysanthemum sensu lato*. 

Furthermore, geographical factors (such as topographic features and river systems) may be involved in hybridization, thereby influencing the distribution patterns of rDNA sites through chromosomal rearrangement (such as breakage inversion, translocation, and recombination) owing to rapid genome changes in the F1 hybrid. First, the plant germplasm (such as the seeds and pollen) is dispersed by the wind and the water. Then, the plant forms the new natural populations or speciation through intra- and interspecies hybridization. For *Isoetes* in East Asia, the spatiotemporal pattern and environmental gradient play essential roles in the speciation and evolution of different ploidy *Isoetes* species from China [51,52]. Similarly, geographical and ecological factors are vital in differentiating and specifying *Chrysanthemum* and its related accessions [53,54]. Moreover, in the *Chrysanthemum* genus, the different ploidy species generally gathered to form hybrids. In the previous studies, the FISH results of the artificial hybridization F1 of *C*. *yoshinaganthum* (2n = 36) × *C*. *vestitum* (2n = 54)*, C*. *indicum* (2n = 36) × *C*. *vestitum* (2n = 54), and *C*. *remotipinum* (2n = 18) × *C*. *chanetii* (2n = 36) showed the site and number of rDNA variations, which also supported the hypothesis above [50]. Combined with the geographic distribution (Figure 2) and distribution patterns of rDNA sites (Figure 3 and Figure 5), we infer that the *Chrysanthemum* accessions are seen as chromosome donors and are located upstream (*C. rhombifolium* (NAU004) and *C. lavandulifolium* var. *aromaticum* (NAU010)). They may have spread through the Yangtze River and may be involved in the karyotype evolution of the downstream accessions *C. indicum* (Wuhan, Hubei) (NAU031) during the evolution process.

### 4.2. Relationship between Chromosome Number and Number of rDNA Sites in Wild Chrysanthemum Genus Plants

Over the past few decades, researchers have found that the number of 5S rDNA sites is related to the ploidy level (chromosome number) in some genera such as cultivated *Cichorium* [55] and *Passiflora* [56]. Conversely, the numbers of 45S rDNA sites show high variability at the same ploidy level (chromosome number) in many different plants [43,57,58]. Significantly, high variability in 45S rDNA sites was also demonstrated in *Anacyclus* (Asteraceae) from other geographical areas at the population level [59].

In the present study, extensive variations in ploidy levels were found in wild *Chrysanthemum* accessions, and ploidy levels varied from diploid (2x) to hexaploid (6x), with a basic chromosome number of 9 (Figure 1 and Figure 3). Compared to the results in previous studies, there were slight differences in the statistical data for rDNA site numbers, especially for the same wild *Chrysanthemum* accessions, such as *C. nankingense* (NAU001) and *C. lavandulifolium* (NAU007). More 45S rDNA sites were discovered in the FISH results by using pTa71(45S rDNA) derived from *T**. aestivum* [11]. Consequently, the non-homologous fragments in rDNA probes may cause the signal noise and interfere with the FISH results. Simultaneously, the 5S rDNA sequences differ in monocotyledon and dicotyledon, and even 5S rDNA is highly conservative in plants. Equally, the FISH results for 5SrDNA sites using homologous probes showed more reliable sites than those using the probes derived from non-homologous species [11,32,33]. From the results of Oligo-FISH analysis, we found that the numbers of 5S rDNA sites were conserved across all ten *Chrysanthemum* accessions and may be related to their ploidy, except for two diploid plants, *C. indicum* (Wuhan, Hubei) (NAU031) (Figure 3a) and *C. mongolicum* (NAU011) (Figure 3d). Furthermore, based on the statistical data of rDNA site number, a significant correlation (*p* < 0.001) existed between the number of 5S rDNA sites and chromosome number, but not between the number of 45S rDNA sites and chromosome number (Figure 4). Similarly, the number of 5S rDNA sites in other genera in the Asteraceae also showed a substantial significant correlation with ploidy (chromosome number), such as *Achyrocline* [60], *Lactuca* [61], and *Centaurea* [62]. In polyploids, the number of duplicated rDNA sites tends to reduce owing to diploidization mechanisms such as multiple chromosomal variations and gene silencing [56]. Furthermore, these mechanisms may further lead to rapid homogenization of rDNA sequences and induce copy number variation, blurring the hybridogenic signatures of allopolyploids [5]. Our findings strongly support the hypothesis above. Consequently, these results provided the possibility of achieving quick ploidy identification based on the number of 5S rDNA sites combined with flow cytometry results.

### 4.3. Potential Mechanism of rDNA Sites Variation

According to previous studies, rDNA sites are the predominant targets of repeated recombination events [63]. Related studies have shown that rDNA arrays and adjacent regions are frequent targets for mobile element insertions [64]. These transposition events might promote changes in the number of rDNA sites. The number and location of 5S rDNA sites are highly conserved in different plants. In this study, we discovered variations in the numbers and locations of 5S rDNA sites in three *Chrysanthemum* accessions: *C. rhombifolium* (NAU004), *C. mongolicum* (NAU011), and *C. indicum* (Wuhan, Hubei, China) (NAU031). These variations may lead to the rapid homogenization of rDNA sequences and induce copy number variation, thereby blurring the hybridogenic signatures of allopolyploids [5]. In addition, the results of other studies indicated that the variation in the number of 5S rDNA sites might be caused by amplification of the covert rDNA copy number during crossing over or transposition events [65]. In contrast, translocation of 5S rDNA sites to other chromosomes without rDNA sites or fusion with different DNA sequences may lead to the disappearance of these sites [20].

Compared with 5S rDNA, 45S rDNA repeats are fragile sites associated with epigenetic alterations and DNA damage under stress responses [16,66]. In addition, high variability in the number of 45S rDNA sites has been observed in many plants [22,67]. Furthermore, we also found that the number and location of 45S rDNA sites varied in some *Chrysanthemum* accessions with the same ploidy (Figure 3 and Figure 5). Therefore, various cytogenetic and molecular mechanisms may underlie the dynamics of the 45 rDNA sites.

First, unequal crossover and transposition events are considered to be the predominant factors that lead to the variation in rDNA sites. Significantly, the entire 45S rDNA repeat sequence in the chromosomes of *Allium* and its subgenus can be transferred from one location to another, suggesting that 45S rDNA may move freely or along with neighboring transposable elements [68]. Regarding conserved 5S rDNA, the different types of 5S rDNA sites that emerged in *C. indicum* (Wuhan, Hubei) (NAU031) may arise from the unequal crossover in homologous chromosomes carrying the same loci. 

Additionally, potential variations in rDNA sites may occur during polyploidy changes to different degrees [69]. Significantly, the different ploidy species within the genera distributed in clusters would contribute to hybridization between intra- and interspecies. These variations may also emerge in the targeted chromosome carrying the rDNA site along with karyotype evolution among the genera, further forming new populations or species. Furthermore, the silencing of rDNA loci in chromosomes would cause the disappearance of the secondary constrictions [70]. Correspondingly, variations in 5SrDNA and 45SrDNA sites were also found between autopolyploid *C. nankingense* (NAU172) and *C. nankingense* (NAU001) (Figure 5). Especially for the variations in 45rDNA sites in autopolyploid *C. nankingense* (NAU172), the potential DNA damage and repair mechanism during the autopolyploid process may cause the 45SrDNA sites of type III to change from long-arm to short-arm. Furthermore, the amplification of Ty1-*copia* and Ty3-*gypsy* elements in long terminal repeat (LTR) retrotransposons along the chromosomes of autopolyploid *C. nankingense* (NAU172) also suggests that the rDNA sites may transfer to another site by these transposition events during autopolyploidization [36]. Repetitive sequences (inverted or tandem) can give rise to the production of dsRNA, an important trigger for RNA silencing as well as heterochromatin formation, and may further lead to diverse phenotypes during the polyploidization [71]. Consequently, the variations in different types of repetitive sequences, such as rDNA sequences, may cause gene expression changes and further lead to abundant phenotypic diversity in the *Chrysanthemum* genus.

Finally, the variation in rDNA sites could also cause high-frequency chromosomal structural fractures and rearrangement, especially for the fragile 45S rDNA sites. Based on the results of previous studies, 45S rDNA is mainly located at chromosomal termini and on satellite chromosomes, which may contribute to the variable number of 45S rDNA sites [66]. Furthermore, the colocalization of 5S and 45S rDNA sites in *C. indicum* (Shennongjia, Hubei) (NAU030), *C. indicum* (Wuhan, Hubei) (NAU031), and *C. indicum* (Tianzhushan, Shaanxi) (NAU047) also strongly supports this potential mechanism of rDNA site variations.

## 5. Conclusions

We studied the molecular cytogenetics of ten wild *Chrysanthemum* accessions using rDNA Oligo-FISH. Based on the Oligo-FISH results, we divided the patterns of 5S and 45S rDNA sites into three types separately and drew idiograms of the rDNA sites of *Chrysanthemum* mitotic metaphase chromosomes. Overall, the 45S rDNA sites showed numerical variations in location and number, whereas the 5S rDNA sites were conserved. In addition, we speculated that clustered groups might greatly promote karyotype evolution in the genus *Chrysanthemum*. Our work may provide a reliable method of examining the sites and number of rDNA variations among *Chrysanthemum* and its related accessions and allow researchers to further understand the evolutionary and phylogenetic relationships of the *Chrysanthemum* genus.

## Figures and Tables

**Figure 1 genes-13-00894-f001:**
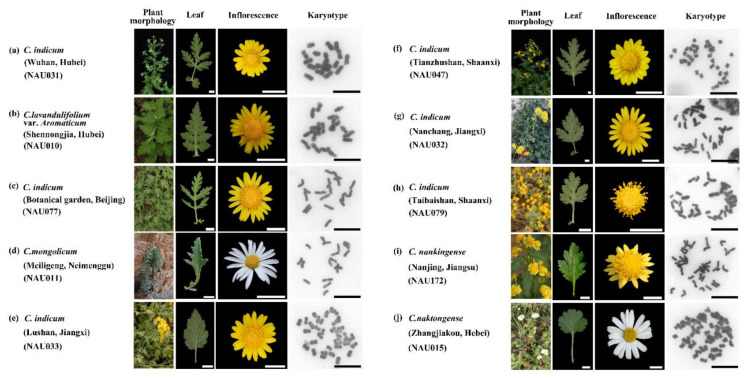
Identification of materials using cytogenetics (black bar = 10 μm) and morphological analysis (white bar = 1 cm): (**a**) *Chrysanthemum indicum* (Wuhan, Hubei) (NAU031); (**b**) *C. lavandulifolium* var. *aromaticum* (NAU010); (**c**) *C. indicum* (Botanical Garden, Beijing) (NAU077); (**d**) *C. mongolicum* (NAU011); (**e**) *C. indicum* (Lushan, Jiangxi) (NAU033); (**f**) *C. indicum* (Tianzhushan, Shaanxi) (NAU047); (**g**) *C. indicum* (Nanchang, Jiangxi) (NAU032); (**h**) *C. indicum* (Taibaishan, Shaanxi) (NAU079); (**i**) *C. nankingense* (NAU172); (**j**) *C. naktongense* (NAU015).

**Figure 2 genes-13-00894-f002:**
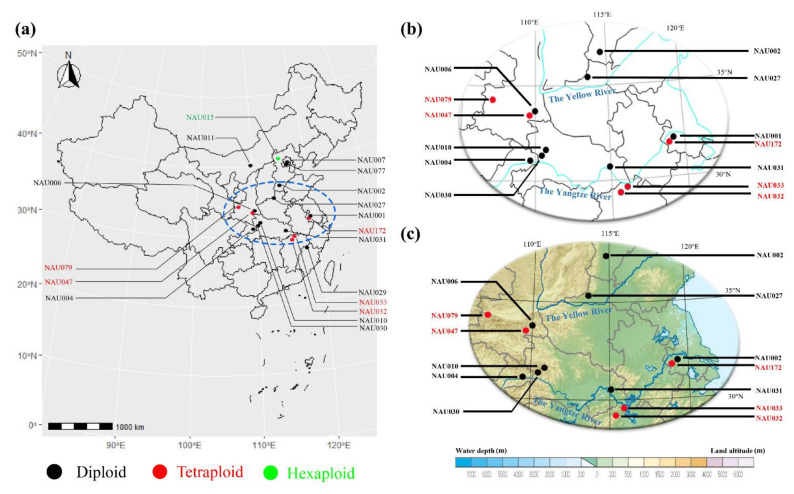
Geographic distribution of related *Chrysanthemum* accessions: (**a**) whole geographic distribution map of *Chrysanthemum* accessions; (**b**) a detailed map of the blue dotted area; (**c**) detailed topography amplitude map of the blue dotted area.

**Figure 3 genes-13-00894-f003:**
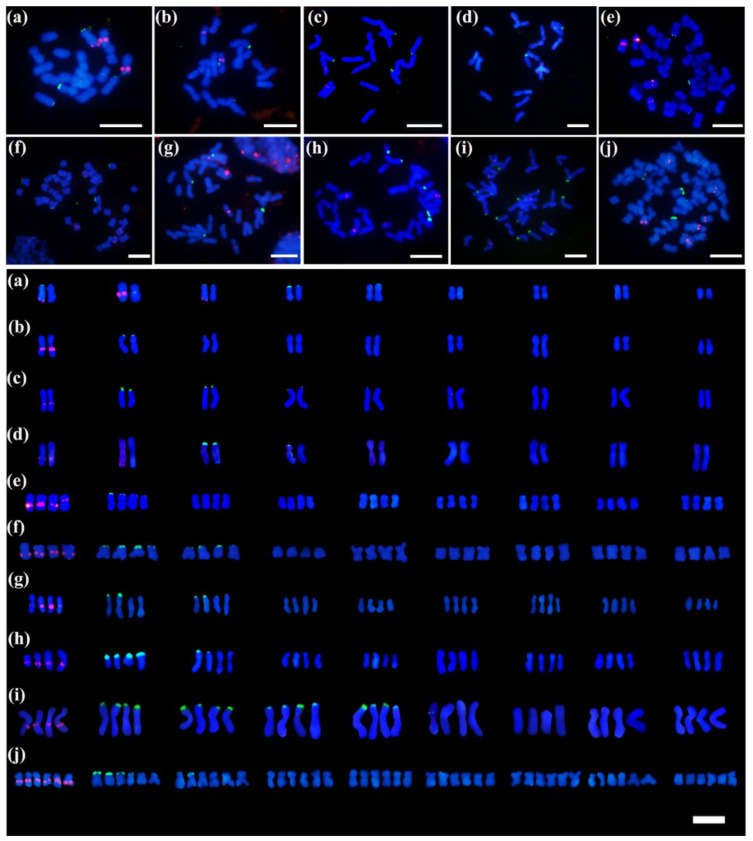
Distribution of 5S (red) and 45S (green) rDNA sites detected by Oligo-FISH: (**a**) *Chrysanthemum indicum* (Wuhan, Hubei) (NAU031); (**b**) *C. lavandulifolium* var. *aromaticum* (NAU010); (**c**) *C. indicum* (Botanical Garden, Beijing) (NAU077); (**d**) *C. mongolicum* (NAU011); (**e**) *C. indicum* (Lushan, Jiangxi) (NAU033); (**f**) *C. indicum* (Tianzhushan, Shaanxi) (NAU047); (**g**) *C. indicum* (Nanchang, Jiangxi) (NAU032); (**h**) *C. indicum* (Taibaishan, Shaanxi) (NAU079); (**i**) *C. nankingense* (NAU172); (**j**) *C. naktongense* (NAU015). Scale bars, 10 µm.

**Figure 4 genes-13-00894-f004:**
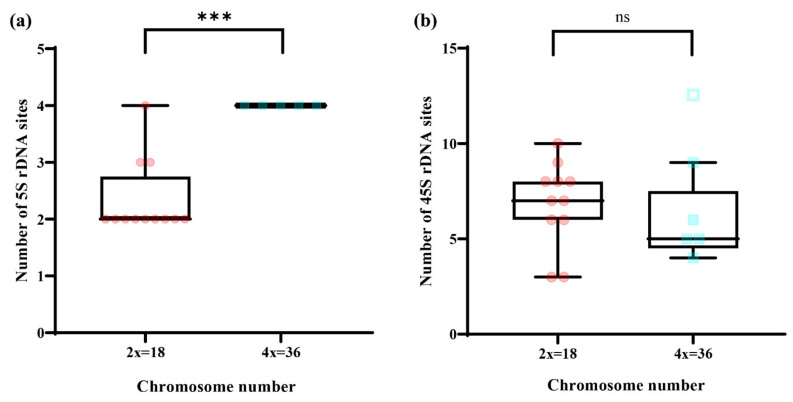
The number of rDNA sites versus the chromosome number (N = 18). The lines show the grouping of the ploidy levels into diploids and tetraploids according to *t*-test: (**a**) *** *p* < 0.001; (**b**) ns *p* > 0.05 (the hollow blue scatter point is the outlier and was excluded in *t*-test).

**Figure 5 genes-13-00894-f005:**
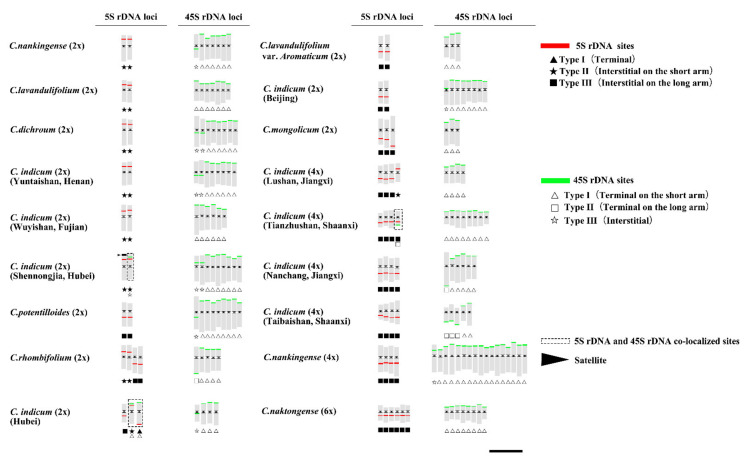
Idiograms of the rDNA sites of *Chrysanthemum* mitotic metaphase chromosomes. The 5S rDNA sites (red), 45S rDNA sites (green), and colocalized sites are marked by boxes with white dotted lines.

**Table 1 genes-13-00894-t001:** Investigated materials with numbers and locations of 5S and 45S rDNA sites in the *Chrysanthemum* genus.

Accession Number	Taxon	Source	No. of 5S rDNA Sites	No. of 45S rDNA Sites	Chromosome Number	Source of Data
NAU001	*Chrysanthemum nankingense*	Nanjing, Jiangsu	2	7	2n = 2x = 18	He et al. [5]
NAU002	*C. dichrum*	Xingtai, Hebei	2	8	2n = 2x = 18	He et al. [5]
NAU004	*C. rhombifolium*	Wushan, Chongqing	4	5	2n = 2x = 18	He et al. [5]
NAU006	*C. potentilloides*	Tianzhushan, Shaanxi	2	9	2n = 2x = 18	He et al. [5]
NAU007	*C. lavandulifolium*	Botanical garden, Beijing	2	7	2n = 2x = 18	He et al. [5]
NAU010	*C.lavandulifolium* var. *a**romaticum*	Shennongjia, Hubei	2	3	2n = 2x = 18	This study
NAU011	*C.mongolicum*	Meiligeng, Neimenggu	3	3	2n = 2x = 18	This study
NAU015	*C.naktongense*	Zhangjiakou, Hebei	6	8	2n = 6x = 54	This study
NAU027	*indicum*	Yuntaishan, Henan	2	8	2n = 2x = 18	He et al. [5]
NAU029	*C. indicum*	Wuyishan, Fujian	2	6	2n = 2x = 18	He et al. [5]
NAU030	*C. indicum*	Shennongjia, Hubei	2	10	2n = 2x = 18	He et al. [5]
NAU031	*C. indicum*	Wuhan, Hubei	3	6	2n = 2x = 18	This study
NAU032	*C. indicum*	Nanchang, Jiangxi	4	6	2n = 4x = 36	This study
NAU033	*C. indicum*	Lushan, Jiangxi	4	4	2n = 4x = 36	This study
NAU047	*C. indicum*	Tianzhushan, Shaanxi	4	9	2n = 4x = 36	This study
NAU077	*C. indicum*	Botanical garden, Beijing	2	8	2n = 2x = 18	This study
NAU079	*C. indicum*	Taibaishan, Shaanxi	4	5	2n = 4x = 36	This study
NAU172	*C. nankingense*	Nanjing, Jiangsu	4	17	2n = 4x = 36	This study

## Data Availability

Data is contained within the article. The figures and tables presented in this study are available here.

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
