# Peer review of "Uneven Levels of 5S and 45S rDNA Site Number and Loci Variations across Wild Chrysanthemum Accessions"

_genes, 2022, doi:10.3390/genes13050894_

Round 1
Reviewer 1 Report
The manuscript submitted for review deals with the analysis of rRNA genes localization on Chrysanthemum chromosomes. The study is an extension of the He et al., 2021 study by adding 10 representatives of this genus demonstrating a wider spectrum of ploidy. The main drawback of the manuscript is the lack of solid scientific content besides stating the fact of the presence/absence of rDNA copies on a particular chromosome.
The given statistics on the relationship between ploidy and the number of DNA copies seem unconvincing (especially fig. 4 the left part). From the data presented, it only follows that diploid chrysanthemums have a variety of localizations of 5S rDNA, while other ploidy does not (and this is not obvious, since there is only one hexaploid representative).
I think that this manuscript (which is technically well done and contains interesting data) could be improved by including a deeper discussion. For example, authors could make some graphical comparison of rDNA localization in Chrysanthemum and other plant genera, demonstrating other patterns (see 255-257 and 294-296). Another idea is to analyze rDNA location diversity in the Chrysanthemum set presented in the context of the phylogeny of this plant set.
In general, the article contains quite interesting material, but rather narrowly and speculatively discussed.
And some remarks:
111-112 some FISH details needed
134-135 do not suggest
142-143 Given from?
145 Most from this study?
149-151 I didn't understand what it was about.
158-159 It is interesting. Needs more discussion, including the functional meaning of 5S and 45S rDNA loci.
223-226 No logic. Do you mean that rDNA is junk DNA?
264-265 Explain, please....
270-272 Explain please and add Fig.2 in the manuscript.
296-297 Which hypotheses?
Author Response
Dear Editors and Reviewers:
Thank you for your letter and for the reviewers` comments concerning our manuscript entitled
“Uneven variations of 5S and 45S rDNA sites in localization and number across wild Chrysanthemum genus accessions” (ID: genes-1661025). Those comments are all valuable and very helpful for revising and improving our paper, as well as the important guiding significance to our research. We have studied the comments carefully and have made corrections which we hope meet with approval. Revised portions are marked in red on the paper. The main corrections in the paper and the responses to the reviewer`s comments are as flowing:
Reviewer1:
1) The study is an extension of the He et al., 2021 study by adding 10 representatives of this genus demonstrating a wider spectrum of ploidy. The main drawback of the manuscript is the lack of solid scientific content besides stating the fact of the presence/absence of rDNA copies on a particular chromosome.
A: Genes encoding ribosomal RNA (rDNA) are universal key constituents of eukaryotic genomes and the nuclear genome harbors hundreds to several thousand copies of each species. One of the best-studied chromosomal landmarks is the ribosomal RNA genes, which usually show considerable variation in number, size, and position among closely related species (Totta et al., 2017). Especially, plant 45S rDNA repeats are fragile sites that are associated with epigenetic alterations and DNA damage. Many environmental stimuli could induce plant nucleolar stress response (Yue et al., 2020). In sharp contrast with polyploids, the evolutionary patterns of rDNA loci numbers in predominantly diploid lineages have received less attention. Consequently, we focus more on the rDNA variations in the site number and location among the wild Chrysanthemum genus to reveal the potential chromosome variation and infer their formation mechanism.
2)The given statistics on the relationship between ploidy and the number of DNA copies seem unconvincing (especially fig. 4 the left part). From the data presented, it only follows that diploid chrysanthemums have a variety of localizations of 5S rDNA, while other ploidy does not (and this is not obvious, since there is only one hexaploid representative).
A: Compared to hexaploid wild chrysanthemums, the diploid and tetraploid wild chrysanthemums account for a large proportion of wild chrysanthemums. In addition, the related studies also showed that the diploid and tetraploid wild chrysanthemums play important roles in the karyotype evolution of the Chrysanthemum genus (Qi et al., 2006; Wang et al., 2019; Yang et al., 2006). Consequently, we conducted the linear correlation analysis mainly based on the diploid and tetraploid wild chrysanthemums because the diploid and tetraploid wild chrysanthemums might be more representative than hexaploid wild chrysanthemums.
3) I think that this manuscript (which is technically well done and contains interesting data) could be improved by including a deeper discussion. For example, authors could make some graphical comparison of rDNA localization in Chrysanthemum and other plant genera, demonstrating other patterns (see 255-257 and 294-296). Another idea is to analyze rDNA location diversity in the Chrysanthemum set presented in the context of the phylogeny of this plant set.
A: Thank you for your constructive comments about this work. Compare with the oligo probes used in this work, the rDNA probe used in other plant genera in Asteraceae such as pTa71(45S rDNA) derived from Triticum aestivum may occur the non-homologous signals and thereby interfere with FISH results. Consequently, based on the FISH result of the homologous and non-homologous probes, the summary analysis may get inaccurate results.
4) 111-112 some FISH details needed
A: We have added the related content in line 112 -122 of the new manuscript.
5) 134-135 do not suggest
A: We have deleted the related content in the new manuscript.
6) 142-143 Given from?
A: We have modified the mistake in line 151 of the new manuscript.
7) 145 Most from this study?
A: We have modified the related content in line 154-161 of the new manuscript.
8) 149-151 I didn't understand what it was about.
A: We have added the explanation in line 154-161 and 279-293 of the new manuscript.
9) 158-159 It is interesting. Needs more discussion, including the functional meaning of 5S and 45S rDNA loci.
A: We have added discussion in line 270-278 of the new manuscript.
10) 223-226 No logic. Do you mean that rDNA is junk DNA?
A: We have rephrased the word in line 232-233.
11) 264-265 Explain, please....
A: We have added the explanation in line 279-293 of the new manuscript.
12) 270-272 Explain please and add Fig.2 in the manuscript.
A: We have added to Figure 2 in the new manuscript.
13) 296-297 Which hypotheses?
A: We have added the related content in line 319-324 of the new manuscript.
References
Qi, S.; Twyford, A.D.; Ding, J.Y.; Borrell, J.S.; Wang, L.Z.; Ma, Y.P.; Wang, N. Natural interploidy hybridization among the key taxa involved in the origin of horticultural chrysanthemums. J Syst Evol 2021.
Totta, C.; Rosato, M.; Ferrer-Gallego, P.; Lucchese, F.; Rosselló, J.A. Temporal frames of 45S rDNA site-number variation in diploid plant lineages: lessons from the rock rose genus Cistus (Cistaceae). Biological Journal of the Linnean Society 2016, 120.
Wang, Y.; Jung, J.A.; Lim, K.-B.; Cabahug, R.A.M.; Hwang, Y.-J. Cytogenetic studies of Chrysanthemum: A Review. Flower Research Journal 2019, 27, 242-253.
Yang, W.; Glover, B.J.; Rao, G.Y.; Yang, J. Molecular evidence for multiple polyploidization and lineage recombination in the Chrysanthemum indicum polyploid complex (Asteraceae). New Phytol 2006, 171, 875-886.
Yue, M.; Gautam, M.; Chen, Z.; Hou, J.; Zheng, X.; Hou, H.; Li, L. Histone acetylation of 45S rDNA correlates with disrupted nucleolar organization during heat stress response in Zea mays L. Physiol Plant 2021, 172, 2079-2089.

Reviewer 2 Report
I have gone through the manuscript titled, " Uneven variations of 5S and 45S rDNA sites in localization and number across wild Chrysanthemum genus accessions. The work is (seemingly) well-structured and well-illustrated, but difficult to understand. The title is very complicated and confusing, I suggest changing the title of the work. You may also want to revise your abstract, trying to highlight your main conclusion. The introduction chapter should be completed and linguistically corrected. I believe that the Materials and Methods section is well structured and scientifically sound. The figures and tables are correct. Literature reviews in the discussion section of the manuscript are good. I feel that editing the text by a native English speaker would help and improve the readability of the paper.
The authors are advised to address the following comments for improving the quality of the article.
Line 30: The current, accepted family name is Asteraceae, Compositae is an old synonym, modify as Asteraceae (Compositae). Please check it through the manuscript.
Line 32: the same problem, change to Asteraceae, delete Compositae (the synonym is above).
Lines 34-41: sentences require rephrasing
Line 60-61: “In the Rosa genus, rDNA probes were used to observe meiotic chromosome behavior and further revealed the evolution of rDNA families in the Rosa genus [23]. - repetitions, sentences require rephrasing.
Line 94-95: “All investigated plant materials were identified using cytogenetics and morphological analysis.” – please specify how many samples there were
Line 96-97: “Morphological identification of leaves and inflorescences was conducted separately during the periods of vegetative growth and blossoming.” - please explain in detail how the identification was carried out which features were taken into account
Line 101. The full name of the genus Chrysanthemum should be added to the heading of Table 1; it is not clear what the abbreviation "C." means.
Lines 131-135 ” Ten wild Chrysanthemum plants were analyzed in this study. Initially, we identified ten wild Chrysanthemum plants by conducting morphological analysis. Chromosome numbers were determined and karyotypes were obtained using cytological observations (Figure 1). These results suggest that the chromosome numbers of the ten wild Chrysanthemum plants ranged (…)” - In general: “ten wild Chrysanthemum plants…” - is enough to write this phrase in one place of the article, preferably in the Materials and Methods section, it does not need to be repeated.
Line 209, 236, C. indicum, Chrysanthemum, 257-258: (…) Brassica, Cucumis, and Artemisia, etc. – in general, Latin taxa names should be italicized. Please check it through the manuscript.
Author Response
Dear Editors and Reviewers:
Thank you for your letter and for the reviewers` comments concerning our manuscript entitled
“Uneven variations of 5S and 45S rDNA sites in localization and number across wild Chrysanthemum genus accessions” (ID: genes-1661025). Those comments are all valuable and very helpful for revising and improving our paper, as well as the important guiding significance to our research. We have studied the comments carefully and have made corrections which we hope meet with approval. Revised portions are marked in red on the paper. The main corrections in the paper and the responses to the reviewer`s comments are as flowing:
Reviewer: 2
1)The title is very complicated and confusing, I suggest changing the title of the work. You may also want to revise your abstract, trying to highlight your main conclusion. The introduction chapter should be completed and linguistically corrected.
A: We have modified these related content in the new manuscript.
2) Line 30: The current, accepted family name is Asteraceae, Compositae is an old synonym, modify as Asteraceae (Compositae). Please check it through the manuscript.
A: We have modified the mistake in line 30 of the new manuscript.
3) Line 32: the same problem, change to Asteraceae, delete Compositae (the synonym is above).
A: We have modified the mistake in line 32 of the new manuscript.
4) Lines 34-41: sentences require rephrasing.
A: We have modified the mistake in line 33-41 of the new manuscript.
5) Line 60-61: “In the Rosa genus, rDNA probes were used to observe meiotic chromosome behavior and further revealed the evolution of rDNA families in the Rosa genus [23]. - repetitions, sentences require rephrasing.
A: We have rephrased the sentences in the line 59-62 of the new manuscript.
6) Line 94-95: “All investigated plant materials were identified using cytogenetics and morphological analysis.” – please specify how many samples there were
A: We have specified the number of the samples in line 95-96 of the new manuscript.
7) Line 96-97: “Morphological identification of leaves and inflorescences was conducted separately during the periods of vegetative growth and blossoming.” - please explain in detail how the identification was carried out which features were taken into account
A: We have added related content in the line 97-99 of the new manuscript.
8) Line 101. The full name of the genus Chrysanthemum should be added to the heading of Table 1; it is not clear what the abbreviation "C." means.
A: We have added to the heading of Table 1 in line 103 of the new manuscript.
9) Lines 131-135 ” Ten wild Chrysanthemum plants were analyzed in this study. Initially, we identified ten wild Chrysanthemum plants by conducting morphological analysis. Chromosome numbers were determined and karyotypes were obtained using cytological observations (Figure 1). These results suggest that the chromosome numbers of the ten wild Chrysanthemum plants ranged (…)” - In general: “ten wild Chrysanthemum plants…” - is enough to write this phrase in one place of the article, preferably in the Materials and Methods section, it does not need to be repeated.
A: We have modified the mistake in line 141-145 of the new manuscript.
10) Line 209, 236, C. indicum, Chrysanthemum, 257-258: (…) Brassica, Cucumis, and Artemisia, etc. – in general, Latin taxa names should be italicized. Please check it through the manuscript.
A: We have modified these mistakes in the new manuscript.

Round 2
Reviewer 1 Report
The manuscript still needs to be corrected. The authors have ignored the most important three first remarks. I bring them back again.
I bring them back again.
1) The study is an extension of the He et al., 2021 study by adding 10 representatives of this genus demonstrating a wider spectrum of ploidy. The main drawback of the manuscript is the lack of solid scientific content besides stating the fact of the presence/absence of rDNA copies on a particular chromosome.
Authors reponse: Genes encoding ribosomal RNA (rDNA) are universal key constituents of eukaryotic genomes and the nuclear genome harbors hundreds to several thousand copies of each species. One of the best-studied chromosomal landmarks is the ribosomal RNA genes, which usually show considerable variation in number, size, and position among closely related species (Totta et al., 2017). Especially, plant 45S rDNA repeats are fragile sites that are associated with epigenetic alterations and DNA damage. Many environmental stimuli could induce plant nucleolar stress response (Yue et al., 2020). In sharp contrast with polyploids, the evolutionary patterns of rDNA loci numbers in predominantly diploid lineages have received less attention. Consequently, we focus more on the rDNA variations in the site number and location among the wild Chrysanthemum genus to reveal the potential chromosome variation and infer their formation mechanism.
There is not even the slightest attempt to add anything substantial. Add please (for example) some information about functional differences between loci analysed, about mechanisms and consequences of their evolution etc. Try (for example) to discuss which evolutionary events in the genome may be associated with variations in the patterns that you describe.
2)The given statistics on the relationship between ploidy and the number of DNA copies seem unconvincing (especially fig. 4 the left part). From the data presented, it only follows that diploid chrysanthemums have a variety of localizations of 5S rDNA, while other ploidy does not (and this is not obvious, since there is only one hexaploid representative).
Authors reponse: Compared to hexaploid wild chrysanthemums, the diploid and tetraploid wild chrysanthemums account for a large proportion of wild chrysanthemums. In addition, the related studies also showed that the diploid and tetraploid wild chrysanthemums play important roles in the karyotype evolution of the Chrysanthemum genus (Qi et al., 2006; Wang et al., 2019; Yang et al., 2006). Consequently, we conducted the linear correlation analysis mainly based on the diploid and tetraploid wild chrysanthemums because the diploid and tetraploid wild chrysanthemums might be more representative than hexaploid wild chrysanthemums.
Linear correlation cannot be obtained from analysis of two points.
3) I think that this manuscript (which is technically well done and contains interesting data) could be improved by including a deeper discussion. For example, authors could make some graphical comparison of rDNA localization in Chrysanthemum and other plant genera, demonstrating other patterns (see 255-257 and 294-296). Another idea is to analyze rDNA location diversity in the Chrysanthemum set presented in the context of the phylogeny of this plant set.
Authors reponse: Thank you for your constructive comments about this work. Compare with the oligo probes used in this work, the rDNA probe used in other plant genera in Asteraceae such as pTa71(45S rDNA) derived from Triticum aestivum may occur the non-homologous signals and thereby interfere with FISH results. Consequently, based on the FISH result of the homologous and non-homologous probes, the summary analysis may get inaccurate results.
However, this probe is designed for the same locus. And you could provide this data with apprppriate comments.
Author Response
Dear Editors and Reviewers:
Thank you for your letter and for the reviewers` comments concerning our manuscript entitled
“Uneven levels of 5S and 45S rDNA sites number and loci variation across wild Chrysanthemum accessions” (ID: genes-1661025). Those comments are all valuable and very helpful for revising and improving our paper, as well as the important guiding significance to our research. We have studied the comments carefully and have made corrections which we hope meet with approval. Revised portions are marked in red on the paper. The main corrections in the paper and the responses to the reviewer`s comments are as flowing:
1) The study is an extension of the He et al., 2021 study by adding 10 representatives of this genus demonstrating a wider spectrum of ploidy. The main drawback of the manuscript is the lack of solid scientific content besides stating the fact of the presence/absence of rDNA copies on a particular chromosome.
Authors reponse: Genes encoding ribosomal RNA (rDNA) are universal key constituents of eukaryotic genomes and the nuclear genome harbors hundreds to several thousand copies of each species. One of the best-studied chromosomal landmarks is the ribosomal RNA genes, which usually show considerable variation in number, size, and position among closely related species (Totta et al., 2017). Especially, plant 45S rDNA repeats are fragile sites that are associated with epigenetic alterations and DNA damage. Many environmental stimuli could induce plant nucleolar stress response (Yue et al., 2020). In sharp contrast with polyploids, the evolutionary patterns of rDNA loci numbers in predominantly diploid lineages have received less attention. Consequently, we focus more on the rDNA variations in the site number and location among the wild Chrysanthemum genus to reveal the potential chromosome variation and infer their formation mechanism.
There is not even the slightest attempt to add anything substantial. Add please (for example) some information about functional differences between loci analysed, about mechanisms and consequences of their evolution etc. Try (for example) to discuss which evolutionary events in the genome may be associated with variations in the patterns that you describe.
A: Thank you for your useful advice. We have added the discussions in line 364-366 and line 372-387 of the manuscript. If it is not suitable, we are willing to continue to modify it
2)The given statistics on the relationship between ploidy and the number of DNA copies seem unconvincing (especially fig. 4 the left part). From the data presented, it only follows that diploid chrysanthemums have a variety of localizations of 5S rDNA, while other ploidy does not (and this is not obvious, since there is only one hexaploid representative).
Authors reponse: Compared to hexaploid wild chrysanthemums, the diploid and tetraploid wild chrysanthemums account for a large proportion of wild chrysanthemums. In addition, the related studies also showed that the diploid and tetraploid wild chrysanthemums play important roles in the karyotype evolution of the Chrysanthemum genus (Qi et al., 2006; Wang et al., 2019; Yang et al., 2006). Consequently, we conducted the linear correlation analysis mainly based on the diploid and tetraploid wild chrysanthemums because the diploid and tetraploid wild chrysanthemums might be more representative than hexaploid wild chrysanthemums.
Linear correlation cannot be obtained from analysis of two points.
A: Compared to hexaploid wild Chrysanthemum accessions, the diploid and tetraploid wild Chrysanthemum accessions account for a large proportion of wild Chrysanthemum germplasm. Consequently, we conducted the t-test analysis mainly based on the diploid and tetraploid wild Chrysanthemum accessions. According to the results, a significant correlation existed between the number of 5S rDNA sites and chromosomes (p < 0.001), but not between that of 45S rDNA sites and chromosome numbers (p > 0.05) (Figure 4). Besides, we added these comments in line 180-186 of the manuscript.
3) I think that this manuscript (which is technically well done and contains interesting data) could be improved by including a deeper discussion. For example, authors could make some graphical comparison of rDNA localization in Chrysanthemum and other plant genera, demonstrating other patterns (see 255-257 and 294-296). Another idea is to analyze rDNA location diversity in the Chrysanthemum set presented in the context of the phylogeny of this plant set.
Authors reponse: Thank you for your constructive comments about this work. Compare with the oligo probes used in this work, the rDNA probe used in other plant genera in Asteraceae such as pTa71(45S rDNA) derived from Triticum aestivum may occur the non-homologous signals and thereby interfere with FISH results. Consequently, based on the FISH result of the homologous and non-homologous probes, the summary analysis may get inaccurate results.
However, this probe is designed for the same locus. And you could provide this data with appropriate comments.
A: Thank you for your useful advice. We have added related comments in line 313-322 of the manuscript.

Reviewer 2 Report
The resubmitted text is ready for publication, there are still some minor errors to correct.
Table 1 and lines 174-175, and line 196 and lines 300-301: "C.lavandulifolium
var. Aromaticum" should be changed to "C. lavandulifolium
var. aromaticum"
Author Response
Dear Editors and Reviewers:
Thank you for your letter and for the reviewers` comments concerning our manuscript entitled
“Uneven variations of 5S and 45S rDNA sites in localization and number across wild Chrysanthemum genus accessions” (ID: genes-1661025). Those comments are all valuable and very helpful for revising and improving our paper, as well as the important guiding significance to our research. We have studied the comments carefully and have made corrections which we hope meet with approval. Revised portions are marked in green on the paper. The main corrections in the paper and the responses to the reviewer`s comments are as flowing:
Reviewer: 2
1) Table 1 and lines 174-175, and line 196 and lines 300-301: "C.lavandulifolium
var. Aromaticum" should be changed to "C. lavandulifolium var. aromaticum"
A: We have modified these related contents in the new manuscript.
